# Association between Cardiac Autonomic Control and Postural Control in Patients with Parkinson’s Disease

**DOI:** 10.3390/ijerph18010249

**Published:** 2020-12-31

**Authors:** Yoan Espinoza-Valdés, Rocio Córdova-Arellano, Maiter Espinoza-Espinoza, Diego Méndez-Alfaro, Juan Pablo Bustamante-Aguirre, Hernán Antonio Maureira-Pareja, Antonio Roberto Zamunér

**Affiliations:** 1Laboratorio de Investigación Clínica en Kinesiología, Department of Kinesiology, Universidad Católica del Maule, 3605 Talca, Chile; yoan.espinoza.valdes@gmail.com (Y.E.-V.); rcordovaarellano@gmail.com (R.C.-A.); maiter.lo.es@gmail.com (M.E.-E.); diegoh.mendeza@gmail.com (D.M.-A.); jpbustamanteaguirre@gmail.com (J.P.B.-A.); 2Laboratorio de Biomecánica y Análisis de Movimiento Humano, Department of Kinesiology, Universidad Católica del Maule, 3605 Talca, Chile; dr.hernanmaureira@gmail.com

**Keywords:** Parkinson’s disease, heart rate, elderly, postural balance, autonomic nervous system, heart rate variability

## Abstract

Parkinson’s disease (PD) is a neurodegenerative disorder that affects postural and cardiac autonomic control. However, since it is unknown whether these changes are associated, the objective of this study was to determine whether such a relationship exists. Twenty-three patients with PD participated. The RR intervals were recorded in different positions and heart rate variability (HRV) was analyzed. Postural sway was analyzed based on the center of pressure. No significant differences on HRV indices were induced by postural change. A correlation was found between these indices and postural control, high frequency (HF), and anterior-posterior (AP) root mean square (RMS-AP) (*r* = 0.422, *p* = 0.045), low frequency (LF)/HF, and AP mean velocity (*r* = 0.478, *p* = 0.021). A correlation was found between HRV induced by postural change and postural control, Δ LF/HF and RMS-AP (*r* = 0.448, *p* = 0.032), Δ LF/HF and ellipse area (*r* = 0.505, *p* = 0.014), Δ LF/HF and AP mean velocity (*r* = −0.531; *p* = 0.009), and Δ LF and AP mean velocity (*r* = −0.424, *p* = 0.044). There is an association between the autonomic and postural systems, such that PD patients with blunted cardiac autonomic function in both the supine and orthostatic positions have worse postural control.

## 1. Introduction

Parkinson’s disease (PD) is a chronic neurodegenerative disorder with a worldwide prevalence of 100–300 cases per 100,000 individuals [1]. It is characterized by a progressive and irreversible loss of neurons in the substantia nigra and a consequent decrease in dopaminergic neurotransmission in the basal ganglia [2].

Clinical changes in PD include rigidity, bradykinesia, tremor at rest, and postural instability [3]. In addition to these motor signs and symptoms, there are non-motor clinical manifestations, such as loss of smell, psychiatric disorders (depression and anxiety), sleep disorders, cognitive dysfunction, and, chiefly, autonomic dysfunction [3]. Among these changes, cardiovascular autonomic dysfunction, which is also known as dysautonomia, should be pointed out [4,5,6,7].

In these patients, autonomic changes can be seen through symptoms, such as orthostatic hypotension, constipation, urinary dysfunction (urgency, retention), sexual dysfunction, excessive sweating, seborrhea, hypersalivation, and sleep and respiratory disorders [8,9]. Among these symptoms, orthostatic hypotension is the most frequent manifestation, with an estimated incidence of 20%–50% [10]. Goldstein [11], retrospectively, studied a group of 35 patients with PD plus orthostatic hypotension and found that 60% manifested orthostatic hypotension at an early stage (within the first year of PD). In four of the patients, orthostatic hypotension preceded PD, while, in four others, orthostatic hypotension was the most important feature of the clinical picture [11]. The consequences of this condition are manifested in locomotion difficulties and decreased mobility, which contributes to an increased risk of falls and decreased independence in activities of daily living, which directly affects the patient’s ability to work and live independently. In fact, the premature unemployment rate due to PD has been shown to range from 27% to 70%. Older age, late illness onset, longer illness duration, greater symptom severity, and lack of support from coworkers are the main factors in work disability [12,13].

This can be explained by the fact that gait and balance control mechanisms are already affected by the disease process, and orthostatic hypotension may further predispose patients to accidental falls and trauma, which leads to increased dependency, fear of falls, and reduced quality of life and may even affect disease prognosis [12,14,15]. Furthermore, autonomic dysfunction increases the patient’s susceptibility to cardiac events since it is associated with high cardiovascular morbidity and mortality. In fact, in patients with PD, cardiovascular events are usually the main cause of mortality [16]. Therefore, identifying cardiac autonomic dysfunction in patients with PD is relevant and could be done by noninvasive methods, such as studying heart rate variability (HRV) [17]. According to Vanderlei et al. [17], HRV is a noninvasive, low-cost, and easy method to assess cardiac autonomic control. Moreover, some studies have shown that HRV indices assessed at rest can be used as prognostic markers for mortality due to cardiovascular events, even in persons with undiagnosed cardiovascular diseases [18,19].

Studies have also shown that there is an association between cardiovascular autonomic control and postural control [15,20,21]. Matinoli et al. [15] studied the association between orthostatic hypotension, mobility, and balance in patients with PD, finding that patients with orthostatic hypotension had greater body oscillation while in a standing position. Similarly, Hohler et al. [20] compared cognitive and motor function in PD patients with and without orthostatic hypotension, finding that patients with orthostatic hypotension had poorer gross motor function, balance, and cognitive function. Thus, although it is well established that the autonomic nervous system plays an important role in postural control, the specific mechanisms underlying this interaction are unclear [21]. Hence, it is relevant to study this interaction and its clinical implications.

Given that autonomic dysfunction can significantly compromise functional mobility, this represents a cardiovascular risk and can exacerbate motor alterations in this population [22]. The objective of the present study was to investigate the association between cardiac autonomic control and postural control in patients with PD. Our hypothesis is that patients with PD who present worse cardiac autonomic control, at rest, and/or in response to an orthostatic stimulus, will present worse postural control. A better understanding of these aspects can help determine whether cardiac autonomic deterioration, even in the absence of orthostatic hypotension, is associated with reduced postural control. If such were the case, cardiovascular interventions could be included in the treatment of non-motor disorders in PD.

## 2. Materials and Methods

### 2.1. Study Population

The sample consisted of 36 male and female volunteers diagnosed with idiopathic PD by a neurologist, according to UK Brain Bank criteria [23,24]. All participants had mild to moderate motor impairment (Hoehn and Yhar scale score 1–3) [23]. They were contacted through community announcements, mainly where PD groups meet and in neurological clinics. Individuals with signs of cognitive impairment according to the Mini-Mental State Examination were excluded [25], as were those with cardiorespiratory and neuromuscular diseases unrelated to PD. Patients were also excluded if they had any acute musculoskeletal condition or musculoskeletal symptoms requiring pharmacological intervention in the last 3 months. The patients’ pharmacological treatments for PD had to remain unchanged for at least 30 days prior to inclusion. After the initial evaluation and familiarization with the experimental procedures, eight participants abandoned the study due to scheduling conflicts, one patient was not diagnosed with idiopathic PD, and the Hoehn & Yhar stage of four patients was >3. Thus, the final sample consisted of 23 individuals. The study protocol was approved by the institutional research ethics committee (protocol #183/2018). After a thorough and detailed explanation of the study’s scope and procedures, all patients provided written informed consent.

Before the experiment, all participants were familiarized with the experimental protocol and were instructed to abstain from stimulants (e.g., coffee, tea, or soft drinks) and alcoholic beverages at least 24 h prior to the examination. Participants were also asked to refrain from strenuous physical activity at least 2 days before the tests. All experiments were carried out in a climate-controlled room (22 to 24 °C) with relative air humidity between 40% and 60%.

### 2.2. Health-Related Quality of Llife

To better characterize the participants, the 39-item Parkinson’s Disease Questionnaire (PDQ-39) was used to assess quality of life. The questionnaire comprises the following eight health dimensions considered to be affected by PD: mobility (10 items), activities of daily living (6 items), emotional well-being (6 items), stigma (4 items), emotional support (3 items), cognition (4 items), communication (3 items), and body movement (3 items). Each question can be rated from 0 to 4. The total score was calculated according to the following formula: 100 × (sum of the items PDQ-39/4 × 39). This percentage was calculated for each of the eight dimensions. Higher scores indicate lower quality of life [26].

### 2.3. RR Interval Recording

All volunteers underwent continuous RR interval recording at a sampling frequency of 1000 Hz (Polar V800 Electro OY, Kempele, Finland). Prior to data collection, the volunteers remained at rest for approximately 20 min to stabilize their heart rate and blood pressure at baseline values. Recordings were performed for 10 min under the following conditions: (1) while resting in the supine position and (2) during active standing. The volunteers were instructed to maintain spontaneous breathing throughout data collection, which was observed and recorded by the examiner through visual inspection of thoraco-abdominal movements.

### 2.4. Heart Rate Variability (HRV) Analysis

Cardiac autonomic control was assessed through HRV analysis in the time and frequency domains. The analysis was performed using Kubios HRV software 3.1.0. Sequences with 256 consecutive RR intervals were selected during the supine and standing periods to perform time and frequency domain analyses. In the time domain, we considered the heart rate and the standard deviation of normal-to-normal intervals (SDNN), which reflects overall HRV [25], and the root mean square of successive differences (RMSSD) in the RR interval, which reflects parasympathetic cardiac autonomic modulation [27]. HRV analysis in the frequency domain was performed by using an autoregressive model. Low frequency (LF, 0.04–0.15 Hz) and high frequency (HF, 0.15–0.4 Hz) bands were quantified in absolute (ms^2^) and normalized units (nu) [27]. Since the LF band is modulated by the cardiac sympathetic and parasympathetic autonomic nervous system (with sympathetic predominance) and the HF band reflects cardiac vagal control, the LF/HF ratio was calculated to assess sympathovagal balance [27], even though its physiological interpretation is controversial [28,29,30].

### 2.5. Stabilometric Assessment

Data on the anterior-posterior and medio-lateral displacement of the center of pressure (COP) were obtained for a period of 60 s using a force platform (Model 9286BA, Kistler Instruments Corp, Winterthur, Switzerland) with a sampling frequency of 100 Hz.

The evaluations were carried out in a quiet room with controlled temperature (22–24 °C) and humidity conditions (40–60%). The participants were instructed to remain in the standing position on a force platform during the recording procedure. To avoid a possible fall hazard, an investigator remained close to the patient throughout the test.

The test consisted of maintaining the standing position for 60 s with the feet parallel (3 cm apart at the heels) and the arms relaxed at the sides. The volunteers were instructed to remain as stable as possible and avoid making any voluntary movements while looking at a fixed point 1 m away. RR intervals were simultaneously recorded for subsequent correlation analysis between cardiovascular variables and postural oscillation. Postural oscillation was analyzed by linear methods using the following indices: ellipse area, root mean square (RMS), and total mean velocity (MV) of the COP [31].

### 2.6. Stabilogram Analysis

The COP data was sampled at 100 Hz for 60 s, from which a series of balance measurements were calculated. The COP RMS distance measures the size of the sway and is believed to be related to the effectiveness or stability achieved by the postural control system [32,33]. Mean velocity (MV) is the mean of the absolute value of COP velocity. Clinically, MV reflects the amount of regulatory activity required to maintain stability [32,34]. The 95% confidence area of the ellipse is expected to cover approximately 95% of the points in the COP path [33]. The ellipse area, which is a method of quantifying the “size” of postural oscillation, is widely used in the literature, having high reliability and validity, in addition to representing two main groups of measurements (position and velocity) [35,36].

### 2.7. Statistical Analysis

The statistical analysis was performed using IBM SPSS Statistics for Windows, vesion 21 (IBM Corp., Armonk, NY, USA). The Shapiro-Wilk test was used to verify normal data distribution. To compare HRV indices between the supine and standing posture, Student’s *t*-test for paired samples was applied for variables with normal distribution and the Wilcoxon test was applied for variables with non-normal distribution. Data are presented as a mean and standard deviation. To determine the magnitude of the difference between the postures, effect sizes were calculated using Cohen’s d and interpreted as a small (from 0.2 to 0.5), moderate (from 0.5 to 0.8), or large effect (≥0.8). The association between HRV indices and COP displacement was verified using the Pearson or Spearman correlation test, depending on data normality. In order to account for the possible confounding effect of the disease severity, a partial correlation analysis between HRV and COP indices was performed controlling for the part III of the Unified Parkinson’s Disease Rating Scale (UPDRS-III) scores. The significance level was set at 5% for all tests.

## 3. Results

### 3.1. Sample Characterization

The demographic and clinical characteristics of the patients with PD are summarized in Table 1.

The scores for the PDQ-39 dimensions are presented in Table 2. The quality-of-life domains with the highest percentage of problems were body movement, emotional well-being, cognition, and activities of daily living. Those with the lowest percentage problems were emotional support, communication, and stigma.

### 3.2. Stabilometric Parameters

The means and standard deviations for COP stabilometric parameters in the standing position were: RMS-anterior-posterior (AP) 4.52 mm (±2.16), RMS-medio-lateral (ML) = 6.22 mm (±3.20), MV-AP = 7.74 mm/s (±5.34), MV-ML = 12.14 mm/s (±5.0), and ellipse area = 480.58 mm^2^ (±287.80).

### 3.3. Heart Rate Variability in Response to a Gravitational Stimulus

Linear indices based on time and frequency of supine HRV and HRV in response to orthostatic stress were evaluated. As shown in Table 3, there were significant differences in the heart rate (HR) (*p* < 0.05) and RR (*p* < 0.05) values, indicating that HR was higher when standing than in the supine position. However, no significant differences (*p* < 0.05) were found for HRV indices between the conditions, which shows that cardiac autonomic modulation had a blunted response to postural change. Figure 1 and Figure 2 depict the individual cardiac autonomic control responses from the supine to standing position. Some participants had an abnormal response to the orthostatic stimulus, characterized by decreases in HR, LFnu, and LF/HF, which should reflect cardiac sympathetic modulation. Other participants had an increase in RMSSD and HF indices, which reflect cardiac parasympathetic modulation.

### 3.4. Relationship between Heart Rate Variability in the Supine Position and Center of Pressure Stabilometric Variables

A correlation analysis was performed between linear indices of supine HRV and COP stabilometric variables. As presented in Table 4, there was a significant positive correlation between HF and RMS-AP (*r* = 0.422, *p* = 0.045, partial correlation: *r* = 0.435, *p* = 0.04), which indicates that greater parasympathetic predominance in the supine position is related to greater instability in the standing position. On the other hand, a significant positive correlation was found between LF/HF and MV-AP (*r* = 0.478, *p* = 0.021, partial correlation: *r* = 0.515, *p* = 0.01), which indicates that greater cardiac sympathetic modulation in the supine position is related to greater effort to maintain control in an upright posture.

### 3.5. Relationship between Postural Variation in Heart Rate Variability and Center of Pressure Stabilometric Variables

A correlation assessment was performed for the delta of linear indices of HRV time and frequency domains in the supine and standing positions (i.e., standing position values minus supine position values) and COP stabilometric variables. Moreover, to avoid a possible confounding effect of the disease severity, a partial correlation controlling for the UPDRS-III was performed. The results are presented in Table 5. A significant positive correlation was found between Δ LF/HF and RMS-AP (*r* = 0.448, *p* = 0.032, partial correlation: *r* = 0.515, *p* = 0.01) and between Δ LF/HF and the ellipse area (*r* = 0.505, *p* = 0.014, partial correlation: *r* = 0.501, *p* = 0.02), which suggests that the exacerbated predominance of the sympathetic component in the standing position is related to greater instability in the standing position. In addition, a significant negative correlation was found between Δ LF/HF and VM-AP (*r* = −0.531, *p* = 0.009, partial correlation: *r* = −0.495, *p* = 0.02), which indicates that greater cardiac sympathetic modulation is related to greater effort to maintain control in an upright posture. On the other hand, there was a significant positive correlation between the index Δ LF(nu) and the ellipse area (*r* = 0.486, *p* = 0.019, partial correlation: *r* = 0.481, *p* = 0.02). Finally, there was a significant negative correlation between Δ LF(nu) and VM-AP (*r* = −0.424, *p* = 0.044, partial correlation: *r* = −0.40, *p* = 0.049).

## 4. Discussion

This study aimed to assess the association between cardiac autonomic control and postural control in patients with PD, in addition to cardiac autonomic modulation to posture change. The main results are discussed below.

### 4.1. Cardiac Autonomic Modulation to Orthostatic Stress

The results of this study showed that PD patients appeared to have a blunted cardiac autonomic response to gravitational stimulation, evidenced by the lack of significant differences in HRV indices (SDNN, RMSSD, LF, HF, LF/HF) and only a moderate effect size between the cardiac autonomic indices obtained in the supine and standing positions. This suggests abnormalities in autonomic response to postural change in PD patients. Furthermore, the reduced increase in the LF/HF ratio in response to a gravitational stimulus suggests a limited increase in cardiac sympathetic modulation to orthostatic stimuli [37]. These findings are in accordance with Barbic et al. [37], who found that, when subjected to a gravitational stimulus, PD patients could not modify cardiac sympathovagal balance toward sympathetic predominance and had a limited increase in the LF/HF ratio. For instance, in the control group of Barbic et al. [37], whose an average age similar to the participants of the present study, the average delta change from supine to tilt was LFnu = 20.5, HFnu = −14.5, and LF/HF = 6.7, while, in the present study, the values were LFnu = 6.51, HFnu = −6.59, and LF/HF = 0.72. Moreover, Alvarado et al. [38] performed an electrocardiogram evaluation upon postural change in patients with PD and found decreased cardiac sympathetic modulation, since the LF index decreased in response to this maneuver. A study by Goldstein [12] on postmortem heart tissue samples from PD patients provides one possible explanation for the present findings. The tissue of four patients had fewer sympathetic axons innervating the left anterior ventricular wall and Lewy bodies in paravertebral sympathetic ganglion cells and axons. It can, thus, be inferred that PD is a pathology of sympathetic denervation, which explains the limited increase in response during orthostatic stimulus.

### 4.2. The Relationship between Sympathovagal Balance in the Supine Position and Postural Control

There was a significant positive correlation between the LF/HF index in the supine position, which reflects sympathovagal balance, and mean COP AP velocity, which suggests that PD patients with greater cardiac sympathetic modulation in the supine position require greater effort to maintain balance in the standing position. Although there should be less modulation of the sympathetic nervous system at rest and greater parasympathetic predominance [39], this relationship is unbalanced in patients with PD due to dysautonomia [40]. Alvarado et al. [38] assessed dysautonomia through HRV, comparing healthy subjects and PD patients in response to a controlled breathing protocol. The results showed that patients with PD had altered vagal modulation, since the RMSSD index was unchanged in response to this maneuver, indicating a probable impairment in the cardiac parasympathetic pathway. These authors also compared cardiac autonomic activity between PD patients and healthy controls [38], finding that patients with PD had greater sympathetic modulation at rest due to an increase in the LF and LF/HF indices of HRV. Regarding postural control, Sibley et al. [21] found that sympathetic discharge is associated with loss of balance. Thus, it is reasonable to assume that the dysautonomia in PD, which is due to cardiac sympathetic denervation, would be related to postural control changes.

### 4.3. Relationship between Cardiac Autonomic Modulation and Postural Control in Response to Postural Change

In our study, analysis of the mean COP velocity and HRV indices (Δ LFun, Δ LF/HF) showed that PD patients with better autonomic modulation to an orthostatic stimulus (greater standing supine delta) had the lowest mean velocity, i.e., those with limited autonomic modulation to postural change required greater regulatory activity to maintain an upright posture. Various studies have suggested that PD patients have a higher mean velocity in a resting position [41,42,43,44]. Our assessment of COP displacement in a standing position yielded similar results to those reported by Rocchi et al. [44], who evaluated COP displacement in a resting standing position in six patients with PD who were off their medication. They found that the mean velocity of patients with PD was higher than normal. Such a result indicates that these patients require a large amount of neural control activity to stabilize the body in a resting position. The authors suggested that the increase in mean velocity is correlated with the tremors the patients suffer. This clinical manifestation is closely related to the pathological progression of the disease, a deficit of dopamine production by the substantia nigra, and dopaminergic denervation of the caudate nuclei and putamen [45]. Thus, it is suggested that impaired postural control is caused by communication changes in the basal ganglia circuit. Studies [46,47,48] have shown that the basal ganglia act as intermediaries between the cerebral cortex and the brainstem, optimizing the selection and execution of the postural response. Evidence also suggests that the basal ganglia are directly implicated in this since both the autonomic and postural pathways share key relay points in the brainstem, cerebral cortex, and basal ganglia [21]. Thus, although it is not possible to determine a causal relationship from the results of the present study, it is reasonable to suppose that the dysautonomia present in PD through the loss of cardiac sympathetic innervation is related to altered postural control due to communication changes between the cerebral cortex and the brainstem, which affects a postural response.

Participants with a higher delta in HRV variables (ΔLF/HF) had a higher RMS, i.e., a closer approach to the stability limit corresponds to better autonomic modulation in response to a postural change. Since this is somewhat contradictory to what is proposed by velocity, these data suggest that anomalous autonomic modulation is related to higher velocity values and lower RMS values. These results can be explained by Maki et al. [49], who draw a distinction between both variables. They suggest that balance activity (velocity) can be related to the degree of stability achieved (RMS). Under certain conditions, an increase in velocity can be observed without a proportional increase in RMS, which may be indicative of compensation for some underlying neuronal or sensorimotor dysfunction, which occurs in PD. On the other hand, it is not clear why patients with the greatest ellipse area and closest approach to the stability limit presented better cardiac autonomic modulation to the orthostatic stimulus, which is identified by lower delta HRV values.

Despite the interesting results, some study limitations should be pointed out. First, since the main objective of the present study was to assess the relationship between cardiac autonomic control and postural control, a healthy control group was not included. Thus, although we have compared our results with some previous published studies, we cannot state categorically that the participants presented a blunted cardiac autonomic adjustment to the orthostatic stimulus. In addition, although no participants presented orthostatic hypotension during the study, future studies should consider using methods such as the Valsalva maneuver, the tilt-table test, or assessing autonomic symptoms through a questionnaire, such as the Composite Autonomic Symptom Score [50]. Such steps could provide more information about the cardiac autonomic profile of these patients.

Another point to be considered is the lack of simultaneous recordings of COP and electromyographic values. A theoretical model called the cardio-postural closed-loop has been proposed and shown interdependence relationships between cardiovascular and postural balance responses [51,52,53]. Briefly, this model addresses the causal relationship between cardiovascular variables with COP oscillations and electromyography activity of the calf muscles, with the last representing the skeletal muscle pump during standing. The authors reported that the skeletal muscle pump mediates the blood pressure responses and may influence the postural control. In addition, muscle pump driving blood pressure oscillations is impaired in the elderly [53]. Although the present study can be considered a start point to understand these interactions, future studies should take into account this approach to clarify the cardio-postural control causal relationship in patients with PD.

Finally, although the linear indices are reliable and several studies have shown their clinical relevance, it is well known that cardiac autonomic control and postural control are both modulated by nonlinear dynamics. Therefore, future studies should consider analyzing the association between nonlinear indices of cardiac autonomic control and COP, as well as integrating electroencephalographic activity, since brain-heart interactions are well known [54] and could play a role in postural control.

## 5. Conclusions

We have found evidence that suggests an association between the autonomic and postural systems, such that patients with greater cardiac autonomic deterioration, both in the supine and standing positions, also present poorer postural control. This implies that interventions in patients with PD should not be restricted to motor alterations, but should also aim at improving the cardiovascular system.

## Figures and Tables

**Figure 1 ijerph-18-00249-f001:**
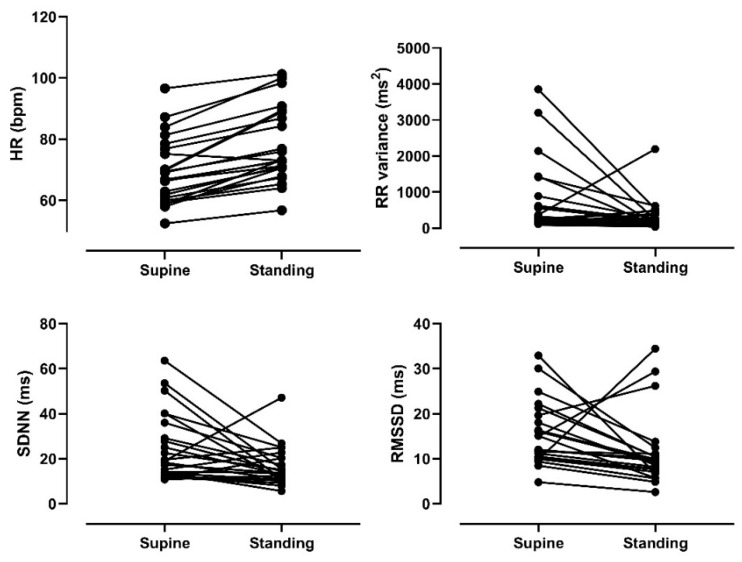
Time domain heart rate variability indices assessed at supine and standing postures. HR: heart rate. SDNN: standard deviation of normal-to-normal RR intervals. RMSSD: root mean square of successive differences in RR interval.

**Figure 2 ijerph-18-00249-f002:**
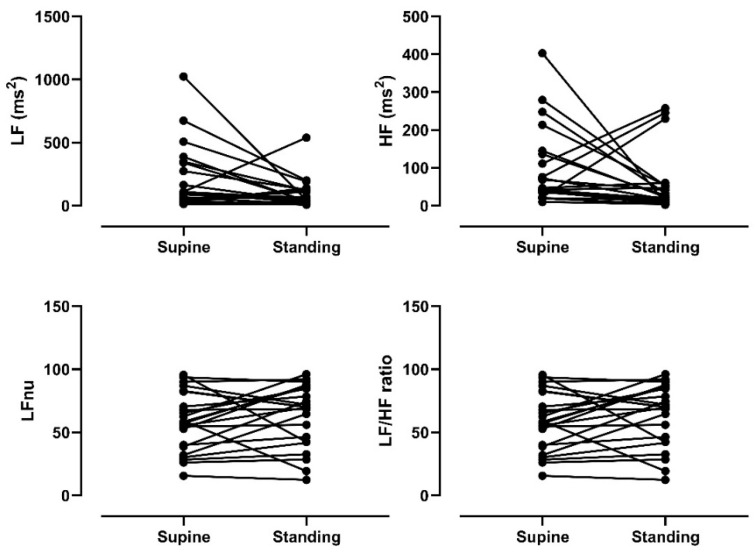
Frequency domain heart rate variability indices assessed at supine and standing postures. HF: high frequency; LF: low frequency; nu: normalized unit.

**Table 1 ijerph-18-00249-t001:** Demographic and clinical characteristics of patients with Parkinson’s disease.

	Mean ± SD
**Age (years)**	68.52 ± 9.26
**Men/Women**	15/8
**Age of PD onset**	60.83 ± 8.52
**Disease duration (years)**	7.70 ± 5.38
**Hoehn & Yahr score (1/2/3)**	7/13/3
**UDPRS-III**	38 ± 16
**LEDD (mg/day)**	383.50 ± 271.01
**Weight (Kg)**	67.07 ± 13.62
**Height (cm)**	159.04 ± 9.66
**BMI (kg/m^2^)**	26.38 ± 4.38

SD: standard deviation. PD: Parkinson’s disease. UDPRS III, Unified Parkinson’s Disease Rating Scale, motor score. LEDD: Levodopa equivalent daily dosage (mg/day).

**Table 2 ijerph-18-00249-t002:** Parkinson’s Disease Questionnaire-39-dimension scores.

	Mean ± SD
PDQ-39 Mobility	40.90 ± 28.14
PDQ-39 Activities of daily living	41.00 ± 22.56
PDQ-39 Emotional Well-being	44.17 ± 26.81
PDQ-39 Stigma	33.50 ± 24.33
PDQ-39 Emotional Support	14.50 ± 23.94
PDQ-39 Cognition	42.75 ± 21.32
PDQ-39 Communication	33.67 ± 29.90
PDQ-39 Body movement	54.00 ± 21.80
PDQ-39 Summary index	38.06 ± 16.78

SD: standard deviation.

**Table 3 ijerph-18-00249-t003:** Indices of heart rate variability evaluated in the supine (Sup) and standing position (Stand).

	Supine	Standing	Delta	Effect Size	Sup vs. Stand *p*-Value
**HR (bpm)**	69.14 ± 10.91	77.80 ± 12.26	8.65 ± 5.01	0.74	<0.001
**SDNN (ms)**	25.15 ± 15.18	16.48 ± 8.86	−7.65 ± 16.28	0.65	0.017
**RMSSD (ms)**	15.55 ± 7.03	11.38 ± 7.89	−1.49 ± 9.13	0.55	0.06
**RR variance (ms^2^)**	750.70 ± 1026.13	317.40 ± 442.27	−433.30 ± 1102.17	0.70	0.07
**LF (ms^2^)**	200.04 ± 252.45	89.47 ± 113.76	−104.38 ± 264.82	0.50	0.06
**LF un**	58.59 ± 23.31	63.96 ± 24.51	6.51 ± 27.26	0.63	0.27
**HF (ms^2^)**	95.17 ± 100.69	52.97 ± 77.78	−29.03 ± 126.10	0.46	0.13
**HF un**	41.32 ± 23.28	35.90 ± 24.45	−6.59 ± 24.17	0.23	0.27
**LF/HF**	3.53 ± 5.28	4.18 ± 5.46	0.72 ± 7.27	0.12	0.67

HF: high frequency. HR: heart rate. LF: low frequency. RR: RR intervals. SDNN: Standard deviation of normal-to-normal intervals. RMSSD: Root mean square of successive differences in RR intervals. nu: normalized unit.

**Table 4 ijerph-18-00249-t004:** Correlation between indices of heart rate variability in the supine position with stabilometric variables of the center of pressure.

	RMS-AP (mm)		RMS-ML (mm)		MV-AP (mm/s)		MV-ML (mm/s)		Ellipse Area (mm^2^)	
	*r*	*p*	*r*	*p*	*r*	*p*	*r*	*p*	*r*	*p*
**HR (bpm)**	0.15	0.50	0.03	0.91	0.25	0.26	0.04	0.86	0.15	0.49
**RR (ms)**	−0.17	0.44	−0.03	0.89	−0.28	0.20	−0.07	0.77	−0.17	0.45
**SDNN (ms)**	−0.16	0.47	−0.21	0.33	0.06	0.77	−0.045	0.84	−0.25	0.24
**RMSSD (ms)**	0.28	0.20	−0.22	0.31	−0.15	0.51	−0.09	0.70	0.02	0.92
**RR variability (ms^2^)**	−0.20	0.36	−0.17	0.45	0.18	0.41	−0.02	0.91	−0.26	0.23
**LF (ms^2^)**	0.03	0.90	−0.15	0.51	0.31	0.15	−0.11	0.62	−0.09	0.69
**LF nu**	−0.28	0.20	−0.06	0.80	0.19	0.39	0.04	0.86	−0.21	0.35
**HF (ms^2^)**	0.42	0.045	−0.13	0.54	−0.19	0.38	−0.21	0.34	0.20	0.36
**HF nu**	0.28	0.20	0.06	0.80	−0.19	0.39	−0.04	0.86	0.21	0.34
**LF/HF**	−0.30	0.16	−0.12	0.57	0.48	0.02	0.06	0.77	−0.32	0.13

HR: heart rate. RR: RR intervals. SDNN: Standard deviation of normal-to-normal intervals. RMSSD: Root mean square of successive differences in RR intervals. LF: low frequency. HF: high frequency. RMS: root mean square. MV: mean velocity. AP: anterior-posterior. ML: medio-lateral. nu: normalized unit.

**Table 5 ijerph-18-00249-t005:** Correlation between the delta from supine to standing of the heart rate variability indices with the center of pressure stabilometric variables.

	RMS-AP (mm)		RMS-ML (mm)		MV-AP (mm/s)		MV-ML (mm/s)		Ellipse Area (mm^2^)	
	*r*	*p*	*r*	*P*	*r*	*p*	*r*	*p*	*r*	*p*
**Δ HR (bpm)**	0.13	0.54	−0.15	0.49	0.40	0.06	−0.14	0.53	0.04	0.85
**Δ RR (ms)**	−0.08	0.72	0.17	0.44	−0.18	0.41	0.14	0.52	0.04	0.87
**Δ SDNN (ms)**	0.14	0.51	0.33	0.12	−0.21	0.34	0.00	0.99	0.29	0.18
**Δ RMSSD (ms)**	0.13	0.55	0.19	0.38	−0.09	0.70	−0.09	0.69	0.14	0.52
**Δ RR variability (ms^2^)**	0.15	0.50	0.22	0.31	−0.24	0.28	0.00	0.10	0.24	0.27
**Δ LF**	−0.06	0.79	0.25	0.26	−0.40	0.06	0.05	0.81	0.12	0.60
**Δ LF nu**	0.28	0.20	0.27	0.21	−0.42	0.04	−0.18	0.40	0.49	0.02
**Δ HF**	−0.32	0.14	0.09	0.70	0.20	0.36	0.24	0.28	−0.26	0.23
**Δ HF** **nu**	−0.35	0.10	−0.31	0.15	0.42	0.045	0.16	0.46	−0.57	0.00
**Δ LF/** **HF**	0.45	0.03	0.14	0.53	−0.53	0.01	−0.23	0.29	0.51	0.01

Δ: delta calculated as difference between values obtained at standing and values obtained at supine. HR: heart rate. RR: RR intervals. SDNN: standard deviation of normal-to-normal intervals. RMSSD: Root mean square of successive differences in RR intervals. LF: low frequency. HF: high frequency. RMS: root mean square. MV: mean velocity. AP: anterior-posterior. ML: medio-lateral.

## Data Availability

The data presented in this study are available on request from the corresponding author. The data are not publicly available due to the consent provided by participants on the use of confidential data.

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
