# Peer review of "Association between Cardiac Autonomic Control and Postural Control in Patients with Parkinson’s Disease"

_ijerph, 2020, doi:10.3390/ijerph18010249_

Round 1
Reviewer 1 Report
(Major concerns)
My major concern is that correlation coefficient across the participants does not mean the relationship between the autonomous function and postural control. Because correlation coefficient is measured across participants, it reflects just severity of the disease. The authors must justify and rebut against this criticism in the text.
The authors conclude that there is an interaction between the autonomic and postural control system as shown in Abstract (Line 24). I disagree with this conclusion. In this study, correlation coefficient was measured across the participants. Correlation across participants does not mean interaction between the two variables. If the authors want to investigate the interaction between the two variables, correlation coefficient must be conducted within a patient.
(Minor concerns)
HR variability
I do not understand why the authors focus on the HR variability in PD patients. Please explain it in Introduction.
L123
In the heart rate analysis section in Methods, detailed explanation of the measurements is missing. Please explain the reason for measuring each parameter.
L218
Authors discussed abnormality based on the present finding. In spite of that, in the present study, measurements in age-matched healthy humans were absent. Please cite any previous findings on the normal participants if you discuss abnormality.
Table 3
Both HR and RR are presented. I think that HR and RR represent same aspect of the physiological status. Please delete one of them.
Term use
The authors frequently use "orthostatic" instead of "standing". The term "orthostatic hypotension" is appropriate, but "orthostatic" may not be appropriate to state the position. In my opinion, "standing" or "in stance" may be better to express the position for using the term with "supine".
The position of the period
In the text, I frequently see a period after brackets citing articles. Probably, in my opinion, placing a period after the brackets is appropriate. Please consider revision.
Author Response
Dear reviewer, we would like to thank you for the time and effort spent reviewing our manuscript. All suggestions and corrections were carefully reviewed in the manuscript and we made all the effort possible to address all the points raised by the four reviewers. We also had our manuscript reviewed by an English native speaker expert in manuscript editing services. We do believe that the current version has been improved.
REVIEWER 1
(Major concerns)
My major concern is that correlation coefficient across the participants does not mean the relationship between the autonomous function and postural control. Because correlation coefficient is measured across participants, it reflects just severity of the disease. The authors must justify and rebut against this criticism in the text.
R: Thank you for this comment. To partially address this point, we performed a partial correlation analysis controlling for the UPDRS score to consider a possible confounding effect of the disease severity on the correlation results. The analysis did not change the results. This was added to the results section (Pages 7 and 9; Topics 3.4 and 3.5).
The authors conclude that there is an interaction between the autonomic and postural control system as shown in Abstract (Line 24). I disagree with this conclusion. In this study, correlation coefficient was measured across the participants. Correlation across participants does not mean interaction between the two variables. If the authors want to investigate the interaction between
the two variables, correlation coefficient must be conducted within a patient.
R: Dear reviewer thank you for this comment. We are not sure if you meant that a repeated measures correlation would be preferred instead. If this is the case, it was not possible since we did not perform a longitudinal study. Even though, we acknowledge that a causal relationship is not possible to determine, however this does not exclude that these variables are related, since both outcomes were collected at the same time in the same participants. To reflect more precisely our results, we changed the term “interaction” by “association”. (abastract and conclusion).
(Minor concerns)
HR variability I do not understand why the authors focus on the HR variability in PD patients. Please explain it in Introduction.
R: A paragraph was added to the introduction to better explain the reason for studying the HRV (Page: 2, 2nd paragraph)
L123
In the heart rate analysis section in Methods, detailed explanation of the measurements is missing. Please explain the reason for measuring each parameter.
R: We added more information in the methods section, topics 2.3 and 2.4 (Page 3).
L218
Authors discussed abnormality based on the present finding. In spite of that, in the present study, measurements in age-matched healthy humans were absent. Please cite any previous findings on the normal participants if you discuss abnormality.
R: We added some data of previous published studies at the discussion (Page 10)
Table 3 Both HR and RR are presented. I think that HR and RR represent same aspect of the physiological status. Please delete one of them.
R: We agree with this comment. In this reviewed version, only HR is presented (See table 2).
Term use
The authors frequently use "orthostatic" instead of "standing". The term "orthostatic hypotension" is appropriate, but "orthostatic" may not be appropriate to state the position. In my opinion, "standing" or "in stance" may be better to express the position for using the term with "supine".
R: Thank you for this observation. We replaced the term orthostatic by standing
throughout the manuscript.
The position of the period In the text, I frequently see a period after brackets citing articles. Probably, in my opinion, placing a period after the brackets is appropriate. Please consider revision.
R: Thank you. This was corrected throughout the manuscript.
Reviewer 2 Report
This manuscript examined the association between cardiovascular function and postural control in Parkinson’s disease (PD) patients and concluded that there is an interaction between the autonomic and postural systems in PD patients. However, there are several concerns regarding the results and the interpretation of this manuscript.
Major concerns
Although, authors described that “All participant had moderate to severe motor impairment (Hoehn and Yahr score 1-3)”, PD patients with H&Y 1-3 scores usually have mild to moderate motor impairment. PD patients with H&Y 4-5 scores have moderate to severe motor impairment. It is also usual that PD patients show clinically relevant cardiovascular autonomic dysfunctions in advanced stage. In PD patients with H&Y 1-3 scores usually have mild or subclinical autonomic dysfunctions. Although, authors described that “PD patients had a blunted cardiac autonomic modulation response to gravitational stimulation, evidenced by the lack of significant difference in HRV indices between the supine and the orthostatic position”, the differences in HRV indices between the supine and the orthostatic position were not trivial and the differences were close to statistically significant (i.e. close to normal condition). HR increased significantly upon orthostatic position, suggesting that autonomic functions were relatively preserved in PD patients recruited in this study.
Furthermore, autonomic functions are basically examined by the degree of autonomic symptoms (light-headedness, syncope) and the orthostatic hypotension, not only by the results of HRV analysis. Unfortunately, the degree of orthostatic hypotension (such as the result of head-up tilt test) was not included in this manuscript.
Another concern is that it is difficult to conclude that the mild abnormalities in HRV analysis are attributable to autonomic dysfunction in PD. For example, age and comorbid conditions such as diabetes mellitus might influence the results of HRV analysis. If authors would like to show that the mild abnormalities in HRV analysis were attributable to PD, the results of age-matched healthy control are necessary.
In addition, it is more difficult to conclude that cardiac autonomic dysfunctions and postural control were associated by only examining the correlational coefficients between the parameters of autonomic functions and postural control. Since, both motor and autonomic functions progress in parallel as disease progress, the correlation between the disease duration and autonomic and postural control should also be examined.
Other comorbid conditions such as cervical and lumber spondylosis might also affect postural control. Although, authors stated that the patients with musculoskeletal diseases unrelated to PD were excluded, comorbid orthopedic conditions such as cervical and lumber spondylosis and knee osteoarthritis are very common in PD patients older than 60 years in my experience. It might be difficult to completely exclude the PD patients with orthopedic comorbid conditions.
Minor points
- The scores of UPDRS (especially part2 and part3), levodopa equivalent dose should be added to Table 1. The dose of dopamine agonist and MAO-B inhibitor which is known to influence the cardiovascular function are also important.
- As I mentioned in “Major concerns”, the degree of orthostatic hypotension (e.g. the results of Schellong test, head-up tilt test) are mandatory to discuss the cardiovascular functions in PD patients.
- The results of the HRV analysis must be interpreted with caution, because the age-matched normal range of HRV parameters are probably not validated.
- The results of PDQ-39 are not discussed with relation to cardiovascular functions and postural control.
Author Response
Dear reviewer, we would like to thank you for the time and effort spent reviewing our manuscript. All suggestions and corrections were carefully reviewed in the manuscript and we made all the effort possible to address all the points raised by the four reviewers. We also had our manuscript reviewed by an English native speaker expert in manuscript editing services. We do believe that the current version has been improved.
REVIEWER 2
1) Although, authors described that “All participant had moderate to severe motor impairment (Hoehn and Yahr score 1-3)”, PD patients with H&Y 1-3 scores usually have mild to moderate motor impairment. PD patients with H&Y 4-5 scores have moderate to severe motor impairment. It is also usual that PD patients show clinically relevant cardiovascular autonomic dysfunctions in advanced stage. In PD patients with H&Y 1-3 scores usually have mild or subclinical autonomic dysfunctions.
R: Thank you for this observation. We corrected this in the manuscript by replacing “moderate to severe” by “mild to moderate” motor impairment (Page: 2; topic 2.1).
2) Although, authors described that “PD patients had a blunted cardiac autonomic modulation response to gravitational stimulation, evidenced by the lack of significant difference in HRV indices between the supine and the orthostatic position”, the differences in HRV indices between the supine and the orthostatic position were not trivial and the differences were close to statistically significant (i.e. close to normal condition). HR increased significantly upon orthostatic
position, suggesting that autonomic functions were relatively preserved in PD patients recruited in this study.
R: Thank you for this observation. We agree with this comment and we reformulated this sentence. We also added 2 figures that illustrate individual responses to the active standing (see Figures 1 and 2), so they show that indeed some participants presented an abnormal response to the orthostatic stimulus. Furthermore, we acknowledge as a study limitation the absence of a control group. (Page: 12; Discussion’s last paragraph).
3) Furthermore, autonomic functions are basically examined by the degree of autonomic symptoms (light-headedness, syncope) and the orthostatic hypotension, not only by the results of HRV analysis. Unfortunately, the degree of orthostatic hypotension (such as the result of headup tilt test) was not included in this manuscript.
R: No participants presented autonomic symptoms nor orthostatic hypotension during this study. We added a sentence as study limitation suggesting that future studies should consider the assessment of autonomic symptoms as assessed, for instance, by the COMPASS 31. (Page: 12; Discussion’s last paragraph).
4) Another concern is that it is difficult to conclude that the mild abnormalities in HRV analysis are attributable to autonomic dysfunction in PD. For example, age and comorbid conditions such as diabetes mellitus might influence the results of HRV analysis. If authors would like to show that the mild abnormalities in HRV analysis were attributable to PD, the results of age-matched healthy control are necessary.
R: We agree with this comment. Indeed, the main objective was to address the relationship between cardiac autonomic control and postural control. So, as mentioned above, we added as study limitation the absence of a control group. (Page: 12; Discussion’s last paragraph).
5) In addition, it is more difficult to conclude that cardiac autonomic dysfunctions and postural control were associated by only examining the correlational coefficients between the parameters of autonomic functions and postural control. Since, both motor and autonomic functions progress in parallel as disease progress, the correlation between the disease duration and autonomic and
postural control should also be examined.
R: Thank you for this comment. To address this point, we performed a partial correlation analysis controlling for the UPDRS score to consider a possible confounding effect of the disease severity on the correlation results. The UPDRS was selected instead of the disease duration because we believe it provides a more objective value regarding the disease severity. This was added to the results section (Pages 7 and 9; Topics 3.4 and 3.5).
6) Other comorbid conditions such as cervical and lumber spondylosis might also affect postural control. Although, authors stated that the patients with musculoskeletal diseases unrelated to PD were excluded, comorbid orthopedic conditions such as cervical and lumber spondylosis and knee osteoarthritis are very common in PD patients older than 60 years in my experience. It might
be difficult to completely exclude the PD patients with orthopedic comorbid conditions.
R: Thank you for this comment. Indeed, it is difficult to rule out all musculoskeletal conditions since the average age was 68 years old. Even though, participants were screened for chronic condition in the initial interview, and no included patients reported to have a formal diagnosis of any chronic musculoskeletal condition. Moreover, we clarified as inclusion criteria, not to have any acute musculoskeletal condition, or to have had musculoskeletal symptoms requiring pharmacological intervention in the last 3 months (Page 2; Topic 2.1).
Minor points
The scores of UPDRS (especially part2 and part3), levodopa equivalent dose should be added to Table 1. The dose of dopamine agonist and MAO-B inhibitor which is known to influence the cardiovascular function are also important.
R: In this study we considered only the Motor domain (part 3) of the UPDRS and it is now included on Table 1. We also added to the Table 1 the LEDD. No patients were using dopamine agonist nor MAO-B inhibitor.
1. As I mentioned in “Major concerns”, the degree of orthostatic hypotension (e.g. the results of Schellong test, head-up tilt test) are mandatory to discuss the cardiovascular functions in PD patients.
R: In this study we focused on the relationship between cardiac autonomic control
and postural control. So, we agree that more autonomic tests would be required to identify cardiovascular autonomic dysfunction. It was acknowledged as study
limitation (Page 12, Discussion’s last paragraph). Even though, the active standing
has proven to show valuable information comparable to the tilt test: https://pubmed.ncbi.nlm.nih.gov/15666065/
https://www.ncbi.nlm.nih.gov/pmc/articles/PMC3812719/
2. The results of the HRV analysis must be interpreted with caution, because the agematched normal range of HRV parameters are probably not validated.
R: We agree with this comment. Indeed, the main objective was to address the
relationship between cardiac autonomic control and postural control. So, as
mentioned above, we added as study limitation the absence of a control group.
(Page 12, Discussion’s last paragraph).
3. The results of PDQ-39 are not discussed with relation to cardiovascular functions and postural control.
R: The PDQ-39 values were presented only to better characterize the participants
in this study. So, it was not the objective of the current study to address its
correlation with the cardiac autonomic control or postural control indices. Even
though, if the reviewer believes it is important to add this as a secondary objective of the study, we could consider performing these correlation analyses.
Reviewer 3 Report
The authors investigated the association between the performance of the postural control system and cardiac autonomic control. More specifically, gold standard parameters of center of pressure during standing and heart rate variability were obtained from twenty three patients with Parkinson's disease. The main results indicated that patients with altered cardiac autonomic function tended to show increased postural sway.
In my opinion, the paper is well written and organized and significantly contributes to the body of knowledge in the field. I would caution the authors to avoid stating that increased measurements of postural sway (as assessed by center of pressure parameters) imply "worse" postural control. As it happens with the cardiac autonomic function, the postural sway might be regulated by a non-linear system. Therefore, increased postural stability is not necessarily "better" and vice versa. English revision is required troughout.
Author Response
Dear reviewer, we would like to thank you for the time and effort spent reviewing our manuscript. All suggestions and corrections were carefully reviewed in the manuscript and we made all the effort possible to address all the points raised by the four reviewers. We also had our manuscript reviewed by an English native speaker expert in manuscript editing services. We do believe that the current version has been improved.
REVIEWER 3
The authors investigated the association between the performance of the postural control system and cardiac autonomic control. More specifically, gold standard parameters of center of pressure during standing and heart rate variability were obtained from twenty-three patients with Parkinson's disease. The main results indicated that patients with altered cardiac autonomic function tended to show increased postural sway.
In my opinion, the paper is well written and organized and significantly contributes to the body of knowledge in the field. I would caution the authors to avoid stating that increased measurements of postural sway (as assessed by center of pressure parameters) imply "worse" postural control.
As it happens with the cardiac autonomic function, the postural sway might be regulated by a nonlinear system. Therefore, increased postural stability is not necessarily "better" and vice versa. English revision is required throughout.
R: We appreciate and agree with the reviewer’s comment. Although the literature reports this interpretation for these indices, it is true that postural sway might be regulated by nonlinear system. So, we added this sentence: “In addition, considering that cardiac autonomic control and postural control are both modulated by nonlinear dynamics, future studies should consider performing non-linear analysis of the center of pressure and of the heart rate variability”. (Page 12, Discussion’s last paragraph).
Reviewer 4 Report
The following study aimed to investigate the association between cardiac autonomic control and postural control in patients with PD. The main finding is an interaction between the autonomic and postural systems, such that PD patients with blunted cardiac autonomic function in both the supine and orthostatic positions have worse postural control. The study has Originality/Novelty, but major concerns must be addressed.
Major concerns
From the authors, the main finding of the current study is the interaction between the autonomic and postural systems. However, the study does not have the methodology required to declare it. HRV and COP collected at the same sampling frequency, isn’t it? No analysis to investigate the coupling or interdependence between variables was performed?
As regards potential limitations of the current study. Does the study have any limitations according to the authors? The absence of EMG recording and analysis could be a major limitation of the current study.
Minor concerns
-The methodology of HRV analysis could be better described. What was the overlap? windowing? time or beats of the analyzed section? Any non-linear index was analyzed?
- In the results section, please see the standard deviation of the following variables (LFn 6.51 ± 27.26 vs. HFn -6.59 ± 24.17). Are there some outliers in the group? I suggest a graph to show inter-individual responses (i.e., responders and non-responders).
- Why would HRV in the supine position have an association with the COP in the orthostatic position? It is not clear whether this is a hypothesis to be tested. Otherwise, the association between cardiac autonomic adjustments (Δ LFun, Δ LF/HF) and postural control responses to orthostatic stress is quite an interest.
- In the discussion section, the authors reported that “Our assessment of COP displacement in a resting position yielded similar results to Rocchi et al., [40]”. Were COP displacements assessed at rest or standing? is it possible to assess the COP at rest (i.e., sitting or supine)?
-Also in the discussion. From the findings of the current study, it is possible to speculate that dysautonomia present in PD is related to the changes between the cerebral cortex and the brainstem, which affects postural response? Would additional approaches, such as the electroencephalogram, be needed?
Author Response
Dear reviewer, we would like to thank you for the time and effort spent reviewing our manuscript. All suggestions and corrections were carefully reviewed in the manuscript and we made all the effort possible to address all the points raised by the four reviewers. We also had our manuscript reviewed by an English native speaker expert in manuscript editing services. We do believe that the current version has been improved.
REVIEWER 4
The following study aimed to investigate the association between cardiac autonomic control and postural control in patients with PD. The main finding is an interaction between the autonomic and postural systems, such that PD patients with blunted cardiac autonomic function in both the supine and orthostatic positions have worse postural control. The study has Originality/Novelty,
but major concerns must be addressed.
Major concerns
From the authors, the main finding of the current study is the interaction between the autonomic and postural systems. However, the study does not have the methodology required to declare it. HRV and COP collected at the same sampling frequency, isn’t it? No analysis to investigate the coupling or interdependence between variables was performed?
R: As mentioned in the methods section, we followed the literature recommendations for the recording of both the COP and RR intervals. So, COP data was recorded at 100 Hz while the RR intervals were recorded at 1000 Hz. Therefore, these variables were not recorded in an integrated mode. This was recognized as a study limitation and proposed as suggestion for future studies (Page 12, Discussion’s last paragraph).
As regards potential limitations of the current study. Does the study have any limitations according to the authors? The absence of EMG recording and analysis could be a major limitation of the current study.
R: We appreciate this comment. This study has several limitations which are now
presented in the discussion section (Page 12, Discussion’s last paragraph).
Minor concerns
-The methodology of HRV analysis could be better described. What was the overlap? windowing?
time or beats of the analyzed section? Any non-linear index was analyzed?
R: The methodology of HRV analysis is now better described. The analysis of the HRV were performed considering 256 consecutive beats. This was added to the methods section (Page 2, topics 2.3 and 2.4). For the analysis we did not use any overlap. In addition, in the current study we did not perform nonlinear analysis, however, we added this sentence: “Finally, although the linear indices are reliable and several studies have shown their clinical relevance, it is well known that cardiac autonomic control and postural control are both modulated by nonlinear dynamics. Therefore, future studies should consider analyzing the association between nonlinear indices of cardiac autonomic control and COP”. (Page 12, Discussion’s last paragraph).
-In the results section, please see the standard deviation of the following variables (LFn 6.51 ±27.26 vs. HFn -6.59 ± 24.17). Are there some outliers in the group? I suggest a graph to show inter-individual responses (i.e., responders and non-responders).
R: Thank you for this suggestion. We added 2 figures to the manuscript.
-Why would HRV in the supine position have an association with the COP in the orthostatic position? It is not clear whether this is a hypothesis to be tested. Otherwise, the association between cardiac autonomic adjustments (Δ LFun, Δ LF/HF) and postural control responses to orthostatic stress is quite an interest.
R: We appreciate this comment. The HRV assessment at supine position was considered since several studies have reported that this measurement as prognostic marker for cardiovascular disease and mortality. A sentence about this was added to the introduction (Page 2, 2nd paragraph)
-In the discussion section, the authors reported that “Our assessment of COP displacement in a resting position yielded similar results to Rocchi et al., [40]”. Were COP displacements assessed at rest or standing? is it possible to assess the COP at rest (i.e., sitting or supine)?
R: Thank you for this observation. The COP cannot be assessed in sitting or supine
postures. So, indeed, this sentence might be misleading. Therefore, we changed to: “Our assessment of COP displacement in a resting standing position yielded similar results to Rocchi et al., [40]”. (Page: 12; Line: 4).
-Also in the discussion. From the findings of the current study, it is possible to speculate that dysautonomia present in PD is related to the changes between the cerebral cortex and the brainstem, which affects postural response? Would additional approaches, such as the electroencephalogram, be needed?
R: Thank you for this interesting comment. We added a sentence about this in the
discussion section and also suggested as perspective for future studies (Page 12,
Discussion’s last paragraph).
Round 2
Reviewer 1 Report
Thank you for your revision.
There is no more comment on this manuscript.
Author Response
I appreciate the reviewer for his/her time and effort in reviewing our study and for the comments which helped us to improve the quality of our manuscript.
Reviewer 2 Report
Although the interpretation of the relationships between the postural control and the results of HRV analysis are complicated, the discussion and the conclusion are plausible.
Author Response

(The authors gave the same response as above.)

Reviewer 4 Report
The study by Espinoza-Valdés and colleagues was improved, but some concerns must be addressed.
1) In the introduction, the hypothesis that was being tested must be included, and the the rationale for this study could be more clarified.
2) As regards the putative mechanisms underlaying hemodynamic influence on COP and EMG oscillations should be addressed in the current study. Please see the following references: Garg et al., 2014 doi: 10.1152/ajpheart.00171.2014 and Rodrigues et al., 2020 doi: 10.1016/j.resp.2020.103384.
3) Despite of several limitations were now included, the absence of EMG is a major limitation that must be consider. Also, R-R and COP signals were not acquired at the same sample rate, so it is difficult to define the causal relationship between these variables. Please consider these limitation in the current text.
Author Response
Dear reviewer, we appreciate your effort and your comments which provided us some new insights and helped us to improve the quality of the current manuscript. Please, find below the reply/corrections to your points. The new corrections in the manuscript were performed using the word track changes, so they are highlighted in red.
Reviewer 4
The study by Espinoza-Valdés and colleagues was improved, but some concerns must be addressed.
1) In the introduction, the hypothesis that was being tested must be included, and the the rationale for this study could be more clarified.
R: The hypothesis is now presented on page 2 (lines 34-36)
2) As regards the putative mechanisms underlaying hemodynamic influence on COP and EMG oscillations should be addressed in the current study. Please see the following references: Garg et al., 2014 doi: 10.1152/ajpheart.00171.2014 and Rodrigues et al., 2020 doi: 10.1016/j.resp.2020.103384.
R: Thank you for suggesting these interesting studies. We included them in the discussion section (Page: 12; Lines 40-50).
3) Despite of several limitations were now included, the absence of EMG is a major limitation that must be consider. Also, R-R and COP signals were not acquired at the same sample rate, so it is difficult to define the causal relationship between these variables. Please consider these limitation in the current text.
R: We appreciate this comment, and we agree with the reviewer. At this study we aimed at assessing the relationship between cardiac autonomic control and COP indices traditionally obtained in the literature and, more important, in clinical settings. However, we agree that in order to clarify possible causal interactions, this suggested approach should be done instead. We added this limitation the discussion section (Page: 12; Lines 40-50).